# Enzymatic Weight Update Algorithm for DNA-Based Molecular Learning

**DOI:** 10.3390/molecules24071409

**Published:** 2019-04-10

**Authors:** Christina Baek, Sang-Woo Lee, Beom-Jin Lee, Dong-Hyun Kwak, Byoung-Tak Zhang

**Affiliations:** 1Interdisciplinary Program in Neuroscience, Seoul National University, Seoul 08826, Korea; dsbaek@bi.snu.ac.kr (C.B.); dhkwak@bi.snu.ac.kr (D.-H.K.); 2School of Computer Science and Engineering, Seoul National University, Seoul 08826, Korea; slee@bi.snu.ac.kr (S.-W.L.); bjlee@bi.snu.ac.kr (B.-J.L.); 3Interdisciplinary Program in Cognitive Science, Seoul National University, Seoul 08826, Korea

**Keywords:** molecular computing, molecular learning, DNA computing, self-organizing systems, pattern classification, machine learning

## Abstract

Recent research in DNA nanotechnology has demonstrated that biological substrates can be used for computing at a molecular level. However, in vitro demonstrations of DNA computations use preprogrammed, rule-based methods which lack the adaptability that may be essential in developing molecular systems that function in dynamic environments. Here, we introduce an in vitro molecular algorithm that ‘learns’ molecular models from training data, opening the possibility of ‘machine learning’ in wet molecular systems. Our algorithm enables enzymatic weight update by targeting internal loop structures in DNA and ensemble learning, based on the hypernetwork model. This novel approach allows massively parallel processing of DNA with enzymes for specific structural selection for learning in an iterative manner. We also introduce an intuitive method of DNA data construction to dramatically reduce the number of unique DNA sequences needed to cover the large search space of feature sets. By combining molecular computing and machine learning the proposed algorithm makes a step closer to developing molecular computing technologies for future access to more intelligent molecular systems.

## 1. Introduction

Molecular computing is a fast-developing interdisciplinary field which uses molecules to perform computations rather than traditional silicon chips. DNA is one such biomolecule which has complementary base pairing properties that allow for both specificity in molecular recognition and self-assembly and for massively parallel reactions which can take place in minute volumes of DNA samples. Pioneering work by Adleman demonstrate solutions to combinatorial problems using molecular computing [1].

Since then, research exploring and exploiting these DNA properties in DNA computing provide the core nanotechnologies required to build DNA devices that are capable of decision-making at a molecular level. Such examples are the implementation of logic gates [2,3,4], storing and retrieving information [5,6], simple computations for differentiating biological information [7], classifying analogue patterns [8], training molecular automatons [9] and even playing games [10]. In particular, molecular computing based on enzymes has also attracted much attention as enzymes can respond to a range of small molecule inputs, have advantage in signal amplification, and are highly specific in recognition capabilities [11].

As DNA computing has the advantage of biocompatibility with living tissue or organisms, implementing molecular computations has also led to promising biomedical applications. An example of this is the logic gates used to specifically target cells or tissue types, thereby minimizing side effects and releasing chemicals at a specific location [12]. Nanostructures have also been used to efficiently deliver cytotoxic drugs to targeted cancerous cells as a therapeutic drug delivery system for various cancers [13,14].

However, these molecular computation approaches were generally preprogrammed, rule-based, or used logic gates for simple forms of computations which may not exceed the ability of reflex action from the perspective of intelligence. Such as in the work of [15,16] where a perceptron algorithm was designed with a weighted sum operation and [17] where a feedforward and recurrent neural network was constructed with cascading nodes using DNA hybridization; although these studies realized pre-defined perceptrons, the idea of learning, where computational weight parameters were updated to train the model was lacking.

Another state-of-the-art molecular pattern recognition work using the winner-take-all model has been recently published, demonstrating molecular recognition using MNIST digit data and DNA-strand-displacement [18]. This work recognizes patterns into defined categories of handwritten digits ‘1’ to ‘9’ using a simple neural network model called the winner-take-all model. Though similar to our study, a key difference is that this work focuses on ’remembering’ patterns during training for recognition and our study focuses on online learning of patterns for classification, where learning refers to the generalization of data following multiple iterations of update during molecular learning. Another key difference is the focus of our work to implement a complete in vitro molecular learning experiment, in wet lab conditions. This is further discussed in the results section as a comparative study with our work (Section 3.4).

Another related area of research includes the implementation of dynamic reaction networks. in vitro biochemical systems, transcriptional circuits have been used to form complex networks by modifying the regulatory and coding sequence domains of DNA templates [19]. A combination of switches with inhibitory and excitatory regulation are used as oscillators similar to that which are found as natural oscillators. Another study also use chemical reactions inspired from living organisms to demonstrate assembling of a de novo chemical oscillator, where the wiring of the corresponding network is encoded in a sequence of DNA templates [20]. These studies use the synthetic systems to further understand the complex chemical reactions found in nature to deepen our understanding of the principle of biological dynamics. A key similarity to our work is the use of modular circuits to model more complex networks. However, it is important to note that these studies are all demonstrated in silico, although it illustrates the potential of in vitro transcriptional circuitry. Computational tools are also being developed, one example being the EXPonential Amplification Reaction (EXPAR), to facilitate the assay design of isothermal nucleic acid amplification [21]. This method helps accelerate DNA assay design, identifying template performance links to specific sequence motifs.

These dynamic system programming paradigms could be valid approaches to implement machine learning algorithms, as programmable chemical synthesis and the instruction strands of DNA dictate which reaction sequence to perform. We ponder that this kind of powerful information-based DNA system technology could also be manipulated to perform defined reactions in specific orders similar to what our study strives to do, thus, implementation operations in vitro to demonstrate molecular learning with the hypernetwork or other machine learning algorithms [22].

Recent work by [23], implement mathematical functions using DNA strand displacement reactions. This study demonstrates considerably more complex mathematical functions to date, can be designed through chemical reaction networks in a systematic manner. It is similar to our work in that it strives to compute complex functions using DNA though a key difference is that the design and validation of this work were presented in silico whereas our work focuses on in vitro implementation of molecular learning. However, the mass-action simulations of the chemical kinetics of DNA strand displacement reactions may be key in developing in vitro learning implementations, as learning consists of target mathematical operations which need to be performed with DNA in a systematic manner to manipulate DNA datasets. Consequently, operations or computational constructs are crucial in implementing machine learning algorithms, from simple perceptrons to neural networks, and this is proposed by this system and thus shares our interests in building systemic molecular implementations of chemical reactions for molecular machine learning. Further examples include a study where an architecture of three autocatalytic amplifiers interacts together to perform computations [24]. The square root, natural logarithm and exponential functions for x in tunable ranges are computed with DNA circuits.

Molecular learning algorithms for pattern recognition in vitro with DNA molecules may be a step towards more advanced systems with higher complexity, adaptability, robustness, and scalability. This could be useful for solving more advanced problems, and be more applicable to use with more intelligent molecular learning devices in order to function in dynamic in vitro and in vivo environments. Some in vitro and in silico studies which aim to create more complex molecular computing systems include associative recall and supervised learning frameworks using strand-displacement [25,26,27,28,29].

There are many difficulties in implementing molecular learning in wet lab experimental settings. DNA may be a more stable biomolecule compared to others such as RNA, however it still requires storage in appropriate solutions and temperature and it is prone to contamination, manipulation techniques often result in heavily reduced yield, and performing and analyzing the molecular biology results can be tedious and time consuming. Furthermore, applying learning algorithms familiar in machine learning bears critical differences to the current demonstration of DNA computing, such as predefined or rule-based models and logic gates.

Our previous study displayed in vitro DNA classification results [30] by retrieving information from a model that was trained with a pseudo-update like operation of increasing the concentration of the matched DNA strands. However, the adaptability and scalability of the model was limited, due in part to the restrictions in the representation of the model by creating single strand DNA (features) with a fixed variable length. Here, this refers to a fixed length of DNA sequence which encodes the variables. Additionally, from the machine learning perspective, updating only with the positive term has critical limitations not common in conventional machine learning methods [31], such as in the log-linear models or neural networks, which require both the positive and the negative terms to perform classification. The accuracy was also somewhat guaranteed because the training and test set were not divided and the features (pool of DNA) were manually designed to have small errors.

In this paper, we introduce a self-improving molecular machine learning algorithm, the hypernetwork [5,27,32], and a novel experimental scheme to implement in vitro molecular machine learning with enzymatic weight update. We present the preliminary in vitro experimental results of proof-of-concept experiments for constructing two types of DNA datasets, the training and test data, with the self-assembling processes of DNA with hybridization and ligation, and DNA cleavage with a nuclease enzyme for the weight update stage of the molecular learning. Our study provides a new method of implementing molecular machine learning with weight update including a negative term.

First, we consider a natural experimental situation typical to machine learning, where we separate the training and test data to evaluate the model. Secondly, by adopting a high order feature extraction method when creating single stranded DNA, a higher-order hypothesis space may be explored which allows for discovery of better solutions even with simple linear classifiers. Thirdly, unlike previous methods which only increased the weight of the model, our proposed method considers both the positive and negative terms of the weight update in the model for learning using an enzymatic weight update operation in vitro. This method is inspired by the energy-based models which use the energy-based objective function to solve problems, and is represented with exponentially proportional probabilities and consists of positive and negative terms to calculate the gradient [33]. Lastly, by encompassing the concept of ensemble learning, the model uses its full feature space for the classification task and also guarantees best performance by voting the best classified labels between each ensemble model. These four aspects are the crucial differences that distinguish our study from previous demonstrations of molecular learning, where without these assumptions of machine learning based aspects, learning, adaptability, and scalability of the model is limited. We show in the results section that the performance of our model gradually increases with the continual addition of training data.

## 2. Materials and Methods

### 2.1. The Molecular Learning Model

The hypernetwork is a graphical model with nodes and connections between two or more nodes called hyperedges (Figure 1 and Figure 2) [27,32]. The connections between these nodes are strengthened or weakened through the process of weight update or error correction during learning [32]. We use the term ‘hypernetwork’ as we refer to hypergraphs which is a generalization of a graph where an edge can join any number of vertices. The hypergraph generally contains nodes and vertices and is a set of non-empty subsets termed hyperedges.

This model was inspired by the idea of in vitro evolution, and provides a clear framework for molecular computing to be realized for molecular learning in a test tube. The probability of hyperedges, or weights, are represented by the concentration of DNA species in the tube. In addition, the idea of weight update is implemented here with specific enzymes, which use the gradient descent in a natural way for autonomous weight update or error correction. Gradient descent is an optimizing procedure commonly used in many machine algorithms to calculate the derivative from the training data before calculating update. All of these reactions occur in a massively parallel manner. Related models have also been previously discussed from the constructive machine learning perspective [34,35].

In this paper, the hypernetwork is interpreted as a maximum entropy classifier with an exponential number of hyperedges as input [36].
(1)$$\begin{array}{c} p\left(y|x;w\right)=\frac{1}{Z}\exp\left(\sum_{i}w_{i}\phi^{\left(i\right)}\left(x,y\right)\right) \end{array}$$
(2)$$\begin{array}{c} \phi^{\left(i\right)}\left(x,y\right)=\delta \left(y,{\tilde{y}}^{\left(i\right)}\right)\prod_{j\in C^{\left(i\right)}}\delta \left(x_{j},{\tilde{x}}_{j}^{\left(i\right)}\right), \end{array}$$
where *Z* is the normalization term, weight $w^{\left(i\right)}$ is the parameter corresponding to $\phi^{\left(i\right)}$, and $C^{\left(i\right)}$ is the set of indices of input features corresponding to *i*th hyperedge. The *i*th hyperedge consists of the set of input variables ${\left\{{\tilde{x}}_{j}^{\left(i\right)}\right\}}_{j\in C^{\left(i\right)}}$, the output variable ${\tilde{y}}^{\left(i\right)}$, and weight $w^{\left(i\right)}$. δ is the identity function. If the whole predefined variables of the *i*th hyperedge are matched to the corresponding variables of an instance, $\phi^{\left(i\right)}$ becomes 1 (Figure 2a).

Our DNA dynamics can be described from the machine learning perspective as presented in Algorithm 1 in Figure 2b. The DNA processes in the paper and Algorithm 1 to the molecular experimental scheme (Figure 3) is matched as following ways:Initializing hyperedge in each epoch corresponds to line 4Figure 3 hybridization corresponds to line 7–8Figure 3 nuclease and amplification corresponds to line 9Merging hyperedges in each epoch corresponds to line 10

In the in vitro implementation of Algorithm 1, the updating of calculated positive or negative term occurs in a slightly different order. In the case of negative weight update, the nuclease cleaves the perfectly matched DNA strands, which occur from the hybridization of complementary DNA hyperedges from training data for ‘6’ and ‘7’, the hybridization being when the negative weight term is calculated. In the case of the positive weight update, the resulting DNA concentrations of each hyperedge from cleavage and purification is amplified, where the positive term was also calculated from the initial hybridization process.

The hybridization rate of DNA datasets to a) construct hyperedges, theta being the hyperdges made from the data and b) to calculate the positive and negative term of Equation (Equation 3), is much faster due to the massively parallel nature of DNA computing, compared to the sequential matching of data in silico (Section 3.1 and Section 3.2 ). DNA data representation through the use of sticky ends and ligation enzyme is almost instantaneous too, due to the use of common complementary strands used to ligate single variable DNA to form free-order hyperedges. This step approximates the kernel function in Equation (Equation 1). The weight of hyperedges in silico is approximated by the relative concentrations of DNA hyperedges, the relative weights of DNA hyperedges being the probabilistic weight calculated in silico, and the updating of weights in Equation (Equation 3) occurs through the PCR amplification and S1 nuclease enzyme cleavage of DNA, thereby increasing and decreasing the concentration of best matched DNA hyperedges respectively (Section 3.3).

The hypernetwork is a suitable model employing molecular machine learning for the following reasons. First, it is a non-linear classifier, which can search the exponential search space of the combination of hyperedges [32,37], unlike the maximum entropy classifier, which has a single variable as input and is a linear classifier.

Secondly, the hypernetwork can be relatively easily implemented in DNA computing as the model utilizes constructive DNA properties such as self assembly and molecular recognition for the generation of hyperedges, and to perform learning operations [30]. Massively-parallel processes can also be exploited with DNA, which means that the search of a large search space is much faster and applicable to the experimental setting.

### 2.2. DNA Dataset Construction

As an example of pattern classification in a test tube, we use the handwritten digit images from the MNIST database [38], which is commonly used to test machine learning algorithms [39]. In our case this is used to test a two-class classification problem with digits ‘6’ and ‘7’ (Figure 4a). The dimensions of the digit images are reduced to 10×10 which are then used as the input data.

From each image 33 pixels are randomly selected in a non-replacement manner to form a hypernetwork model. 33 pixels were used as we wanted to use the least amount of pixels to represent the largest search space of 10 by 10 pixel images. In other words, only 33 pixels were used in each ensemble of each iteration. So 33 unique DNA sequences could be used to represent 33×3, so 99 pixels in total for each ensemble of learning. In our experiment, we produce three ensembles for each image (Figure 5). The 33 unique pixels from each ensemble are encoded to DNA by allocating 33 unique DNA sequences consisting of 15 base pairs each. Unique DNA oligomers are sequenced with an exhaustive DNA sequence design algorithm, EGNAS [40]. EGNAS, stands for Exhaustive Generation of Nucleic Acid Sequence, and is a software tool used to control both interstrand and intrastrand properties of DNA to generate sets with maximum number of sequence designs with defined properties such as the guanine-cytosine content. This tool is available online for noncommercial use at http://www.chm.tu-dresden.de/pc6/EGNAS. Once each DNA oligomer is assigned to a pixel, that is, one unique DNA to each pixel (Figure 4b), it is the grey scale value (between 0 and 1) for each pixel that determines the amount of DNA to be added to make the DNA dataset (Figure 4a). Each class is labeled with a different fluorescent protein to allow visualization of classes, allowing for the learning of two classes in one tube.

Following the addition of relative amounts of DNA oligomers according to the pixel value, the sequences are joined together to produce free-order hyperedges for the initialized hypernetwork, and training and validation datasets (Figure 4b,c). Here, free-order refers to any number of linked variables in the DNA sequence, for example, 1-order hyperedge consists of one variable, 2-order hyperedge with two variables and 3-order hyperedge with three variables and so on (Figure 3c). Using PCR, the variable DNA, in this instance the pixel, the forward and reverse primers are hybridized to their respective complementary strands to form double stranded DNA. These three units act as building blocks for constructing free-order hyperedges as they are annealed at the tag or sticky end regions with enzyme ligase. It is worth noting that free-order hyperedges enhance the robustness of the model, and it is not only the variables that are learned through the self-organizing hypernetwork, but also the order of hyperedges.

### 2.3. Learning with Enzymatic Weight Update

Our main idea is that the dual hybridization-separation-amplification process with enzymatic weight update can be interpreted as an approximation of the stochastic gradient descent of hypernetworks.

In the test tube, enzymatic weight update is realized with enzymes which target specific DNA structures. Molecular recognition through hybridization of complementary base pairs allows matching of data to form symmetrical internal loops if incorrectly matched. Symmetric internal loops of DNA [41,42,43] are used to correlate the differences in training instances. This physical DNA structure is used to determine the degree of matching between two complementary strands for pattern matching. It is these DNA structures which are cleaved by specially chosen enzymes to perform the enzymatic weight update stage of the learning process. Consequently, the cleaving results in decrease in concentration of DNA with symmetric internal loops in the test tube.

First, the training data for digits ‘6’ and ‘7’ is hybridized with the training data for ‘7’ (Figure 3a). Then, S1 nuclease, an enzyme which cleaves the perfectly matched DNA sequences (completely hybridized hyperedges) is added. This allows for the selection of only the perfectly matched hyperedges, leaving any degree of mismatched hyperedges in the tube. This is the enzymatic weight update operation that is demonstrated in vitro. DNA is then purified, separated using biotin, and amplified with PCR resulting in a mixture of single-stranded DNA hyperedges exclusive to ‘6’. This is repeated for the training data for ‘6’ to train ‘7’ (Figure 3b).

With the enzymatic weight update, through the decrease weight function of S1 nuclease, we eliminate the hyperedges common to both ‘6’ and ‘7’ resulting in only exclusive hyperedges characterizing ‘6’ and ‘7’ for successful digit classification. This corresponds to the negative weight update idea. The discrepancy between the two classes of digit data is represented by the remaining pool of DNA sequences, which are then added to one tube. The addition of the remaining sequences symbolizes the positive weight function. The trained hypernetwork from mini-batch 1 is added to the next minibatch and so on. Here the concept of online learning is applied. The weak learner 1, 2, 3 for ensembles 1, 2, and 3 respectively is added at each iteration to form the final trained weak learner after mini-batch 5 (Figure 5). The ensemble method is discussed in the next section. Repetition of these learning steps is predicted to construct an ensemble of three molecular classifiers which can be used for ensemble prediction in the test stage by measuring the final ratio of fluorescence for Cy3 (Label for 6) and Cy5 (Label for 7). To perform online learning, after the model is trained with each mini-batch, the DNA pool is combined to create a final hypernetwork model for given classification tasks.

The above process can be described theoretically, where the set of hyperedges is determined or the connections between the nodes are strengthened or weakened through the process of weight update or error correction during learning. Equation (Equation 3) is the gradient of the log-likelihood of Equation (Equation 1).
(3)$$\Delta w_{i}=\frac{1}{N}\sum_{n=1}^{N}\phi^{\left(i\right)}\left(x^{\left(n\right)},y^{\left(n\right)}\right)\left[1-\frac{1}{Z}\exp\left(\sum_{i^{\prime }}w_{i^{\prime }}\phi^{\left(i^{\prime }\right)}\left(x^{\left(n\right)},y^{\left(n\right)}\right)\right)\right]$$

The next step of the algorithm consists of pattern matching and weight update (Algorithm 1, line 7, 8). Equation (Equation 3) shows the gradient of the log-likelihood of Equation (Equation 1). Our algorithm illustrates our learning process of hypernetworks, which can be naturally applied to online learning. The algorithm consists of both parameter learning and structure learning. In parameter learning, Equation (Equation 3) is used for stochastic gradient descent. Equation (Equation 3) consists of the positive term $\phi^{\left(i\right)}\left(x,y\right)$ and the negative term $-\frac{1}{Z}\cdot \phi^{\left(i\right)}\left(x,y\right)\cdot exp\left(\sum_{i^{\prime }}w_{i^{\prime }}\phi^{\left(i^{\prime }\right)}\left(x,y\right)\right)$.

Without the terms of matching instance $\phi^{\left(i\right)}\left(x,y\right)$, the positive term is 1 and the negative term is $\frac{1}{Z}\cdot \phi^{\left(i\right)}\left(x,y\right)\cdot exp\left(\sum_{i^{\prime }}w_{i^{\prime }}\phi^{\left(i^{\prime }\right)}\left(x,y\right)\right)$. In structure learning, the feature set of hyperedges Φ is updated. The number of possible kernel functions $\phi^{\left(i\right)}$ is exponential to the order of the hyperedge. This is required to select the subset of hyperedges, which consist of separable patterns. The candidate hyperedges are sampled from the data instance, where values of the partial input of an instance are used as the features. The hyperedges which are not important are pruned. Large absolute weight values or non-negative weight values can be used as the measure of importance; we use the latter case.

### 2.4. Ensemble Learning of Hypernetworks

Figure 5 shows that three ensembles were used to train images ‘6’ and ‘7’ in an online manner. We apply the ensemble method to our model for the following reasons. First, the ensemble method guarantees maximum performance within the ensemble models. When different types of models are being ensembled together, the overall performance increases as each model’s characteristics of representation and search spaces are different from one another [44]. Another reason for creating a three ensemble model is due to the limitation of interpretability. Since our designed model created free-ordered hyperedges, using 100 pixel produces 100! (factorial of 100) different hyperedges which is almost impossible to visualize with current electrophoresis techniques. Moreover, the formation of the hyperedges is unpredictable since it is affected by a range of external stimuli (temperature, time, concentration etc.). Therefore, we divide the image into three sets and perform the voting method with the final produced results by each of three ensemble models.

The inference procedure is almost the same as the learning process. However, before the S1 nuclease is applied, the concentration of the perfectly matched DNA is measured to decide whether the test data is a digit ‘6’ or ‘7’.

## 3. Results

To ensure that the experimental protocol is implemented to the highest degree of efficiency and accuracy, a series of preliminary experiments were undertaken to validate the experimental steps involved in demonstrating the molecular hypernetwork.

### 3.1. DNA Quantification of 3-Order Hyperedges

The formation of a random single-stranded library was critical in verifying the success of the full experimental scheme. The experimental steps and results are as follows:Hybridization of upper and bottom strands of variable units.Ligation of these variables in a random fashion, all in one tube to create a double-stranded DNA random library.Purification of the sample from ligase.Separation of the double-stranded library to a single-stranded random library using Streptavidin and Biotin.Centrifugal filtering of DNA for concentration.Verification of library formation with the use of complementary strands.

Each step listed above was carried out using wet DNA in a test tube, and at each step, the DNA concentration was measured using a NanoDrop Nucleic Acid Quantification machine, which is a common lab spectrophotometer (Figure 6). Hybridization was performed on the PCR machine with a decrement of 2° from 95° to 10° 100 pmol of each upper and lower strand was used.

Ligation was carried out using Thermo Fisher’s T4 ligase enzyme. This enzyme joins DNA fragments with staggered end and blunt ends and repairs nicks in double stranded DNA with 3’-OH and 5’-P ends. Three units of T4 ligase was used with 1 μL of ligation buffer.

Annealing of the ligated DNA strands are then put into PCR conditions with a decrement of 1 degree from 30 to 4 degrees. Purification and extraction of DNA from the ligase inclusive sample was carried out using the QIAEX II protocol for desalting and concentrating DNA solutions. The standard procedures for this were used [45,46,47]. This procedure is commonly used to remove impurities (phosphatases, endonucleases, dNTPs etc.) from DNA, and to concentrate DNA. The QIAEX desalting and concentration protocol gives quite a detailed description of the procedure.

While there was a significant loss of DNA content after ligation, a sufficient concentration was recovered from the centrifugal filtering step allowing identification of the nine complementary strands possible from the combinations of the three different variables initially used. Bands at the 70 bp marker were present for all nine types of sequences which confirm that all possible sequences were successfully constructed and retrieved during the experimental process. The results shown in Figure 6 present DNA concentrations at various stages of the learning process, and the final confirmation of the success in making a random double-stranded library.

### 3.2. Creating Free-Order DNA Hypernetworks

For the creation of the free-order hyperedges or different lengths of hyperedges, the concentrations of DNA sequences according to its corresponding pixel greyscale value, was added to a tube and ligation performed as described above. The PAGE electrophoresis gel illustrates the free-order hyperedges which were produced from the ligation procedure (Figure 4c and Figure 7a). Against the 10 bp ladder, many bands of varying intensities can be seen, representing different lengths of DNA present in the sample. From the original tube of 15 bp single stranded DNA following the ligation protocol, the formation of random hyperedges from 30 bp to 300 bp are visible (Figure 7a). This correlates to 0-order to 15-order hyperedges. The same procedure was carried out to produce free-order hyperedges for every dataset: Training data and test data.

### 3.3. Weight Update Feedback Loop for DNA Hypernetwork

Figure 7b shows the result of enzyme treatment for S1 nuclease. DNA was incubated with the enzyme for 30 min at room temperature than enzyme inactivated with 0.2 M of EDTA at heating at 70° for 10 min. The control lane shows 4 bands, the perfectly matched strand, 1-mismatched, 2-mismatched, and 3-mismatched strands from the bottom to the top of the gel. The function of the S1 nuclease is investigated for decreasing weights or in this case DNA concentration, where perfectly matched DNA sequences are cleaved and mismatched sequences remain. In the 0.5 S1 lane, it is evident that only the perfectly matched DNA strands are cleaved and the mismatched strands remain in the mixture. This represents the sequences only present exclusively in the data for digits ‘6’ or ‘7’ can be reproduced with the use of S1 nuclease in the weight update algorithm.

It is interesting to note that the issue of scalability may be addressed through our design which allows 10-class digit classification within the same number of experimental steps. More classes of training data could be added in the hybridization stage to all but the one class of training data for which the label is being learned. This provides a larger scale of digit classification without drastically increasing the workload, time or the need to order new sequences. This novel method of implementing digit classification and experimental results demonstrate the enzymatic reactions which is prerequisite to making this experimentally plausible.

### 3.4. Performance of In Silico Experiment of DNA Hypernetwork

As described in the Materials and Methods section, we used the MNIST dataset to measure the classifier accuracy [48], which is defined as the estimation of number of correctly classified objects over the total number of objects.

To verify the learning capability of our proposed model, both incremental and online aspects, we compared our model to two existing models; the perceptron model described in [18] and the conventional neural network [49] as a representative example of non-linear classification.

In [18], a basic perceptron model outputs the weighted sum for each class and selects the maximum value as their winning final output. 2-class classification between digits ‘6’ and ‘7’ is demonstrated and nine label 3 grouped class-classifier is described, where all methods first eliminate the outlier and the performance achieved by providing probabilistically calculated weights of the 10 most characteristic features to the designer as a prerequisite. However, in our study, we do not eliminate outliers or give prior weights and use the MNIST dataset as it is for our performance. We exploit the learning ability of a DNA computing model without the need for the designer to previously define weights. Not only do we reduce the labor required by the designer to define weights for selected features but we exploit the massively parallel processing capability of DNA computing whilst demonstrating molecular learning which improves performance with experience as our model is designed for implementation in vitro through molecular biology technique with wet DNA.

Two types of initialization of weights are introduced in our simulation results, 0 weight initialization which is easily implemented in DNA experiments, and random weight initialization which is harder to be conducted in vitro but is more conventionally used in perceptron and neural network models. The perceptron and neural networks convergence of performance are dependent on their initialized weights [50]. We conducted these two methods of initializing the weights, first starting with 0 weight, and second providing random values to the weights.

For the 2-class classification, 1127 and 11,184 images were used for the test and training data respectively. For the 10-class classification 10,000 and 60,000 images were used for the test and training data respectively. As the MNIST dataset is balanced over all classes and not skewed to any class, the accuracy measurement is sufficient to evaluate the classifier’s performance [51]. For all cases of learning, we randomly sampled five images. We did this to demonstrate our model’s capacity to implement online learning in only a few iterations and more significantly for our work, for the correlation to our wet lab molecular learning protocol, where only five iterations of molecular learning experiments need to be carried out for learning to produce classification performance.

For both the perceptron and neural network model, the learning rate was set to 0.1. For the perceptron model, of the 10 output values from the given input, the output with the biggest value was selected in a winner-take-all method. As the hyperedges produced from the hypernetwork in the 10-class classification (all with weights) was 284, we chose to use 300 hidden nodes for the neural network. All the source codes and relevant results can be found on github repository (https://github.com/drafity/dnaHN).

Figure 8 shows the results of our in silico classification results. As the number of epochs increased, the test accuracy of the hypernetwork also increased. The accuracy of the hypernetwork was also higher than the comparative models of the perceptron and neural network. We note here the significance of having used an accuracy measure to evaluate our classifier. The DNA learning models implemented in vitro in the mentioned related works often lacked appropriate measures to evaluate the classifier’s performance. Furthermore, though recognition abilities have been reported through in vitro molecular learning, classification of data through learning and testing this, to consequently group or label the unknown test data to a category by a molecular learning model is to the authors’ knowledge a novelty in itself.

A key feature of our model in comparison to the perceptron or neural network models is the minimum number of iterations required to observe significant performance. Our proposed model only needs five iterations of learning to achieve significant classification performance. However, as the results show in Figure 8 and Figure 9, initializing the weight to 0 or giving random weights to the compared models still resulted in low accuracy in small epoch sizes. The perceptron and the neural network require a much larger epoch size for significant classification performance to be achieved.

This is crucial as in vitro experiments to perform molecular learning not only require time-consuming laborious work, but issues with contamination and denaturation can affect the quality of the experimental results. It is only more suitable for molecular learning experiments performed in wet lab conditions to be efficient, exploiting the massively parallel computing possible with DNA but also minimizing the protocol required to perform molecular learning. Our model is designed to do this by autonomously constructing higher order representations, using massively parallel DNA processes to create and update weights in minimal iterations. Furthermore, compared to state-of-the-art studies in molecular recognition, we were able to achieve over 90% accuracy and 60% accuracy in 2-class and 10-class classification respectively, through a molecular learning algorithm in five iterations. Thus, this result present that our model is a novel molecular learning model which learns in an online manner through minimal iterations of learning, suitable for wet lab implementations using DNA.

The hypernetwork, inspired by DNA chemical reactions, when computed in silico, clearly showed the disadvantage of sequential computing in silico and the massively parallel processing advantage of DNA computing in vitro. In an instant, DNA molecules hybridize when complementary strands are added together in an appropriate buffer and thus almost immediately the computing in that tube comes to an end. However, implementation of the hypernetwork in silico is iterative, sequential. For each training and test data, the number of matches and mismatches need to be calculated sequentially, and as the order of hyperedges increases, computational time complexity increase exponentially. As a result, with our computing power, empirically, 1000 iterations require 1000×20 min, a total of approximately 10 days to complete. Therefore it is important to note that there is a sheer advantage in DNA implementation of the hypernetwork compared to in silico. For the same reason, the neural network requires around 1000 iterations to converge and in the case of non-linearly separable data when using the perceptron model, it fails to converge. Thus, the proposed hypernetwork may also be introducing the possibility of a new computing method.

We also discuss the reasons why the perceptron does not perform as well. Due to the nature of perceptron models, as a representative linear classifier, it is difficult for it to solve linearly inseparable problems (XOR problems) without any preprocessing or adding layers to the perceptron to deal with non-linearity problems [52,53]. As illustrated in Figure 8, the perceptron model shows performances close to what would be achieved from random picking for both small and large number of iterations. Depending on how the data is fed, 2-class classification performance levels show major fluctuations, where up to 80% performance is achieved at times and others much lower performance. This phenomenon is typically representative of unsuccessful generalization of the data also called overfitting. For example, in the case of the perceptron, as described in the reference, the performance is achieved only for the data that can be fitted linearly. To learn linearly inseparable data, the model needs a feature reduction or extraction preprocessing methods [54] or a nonlinear kernel to model (e.g., Support Vector Machine [55], Neural Network [53] the high dimension input dataset. As this paper focuses on the implementation of a learning model in vitro only using easy, basic and fundamental learning processes, we believe this is out of the scope of our paper and omit further discussion.

As a support to such arguments, as shown in Figure 8 and Figure 9, both in our results and in Cherry and Qian’s work, there are cases where a variety of elimination conditions and previously providing the optimal weights of batch data by the designer can achieve significant performance in 2-class classification i.e., overfitted results (the maximum value of the error bars). However, as in the case of 10-class classification tasks, where the data is not linearly separable where it exceeds the model’s capacity, the range of performance levels are smaller and, as acknowledged by Cherry and Qian in their paper, it is difficult for the designer to find the optimal weights for the model.

## 4. Discussion

We have proposed a novel molecular machine learning algorithm with a validated experimental scheme for in vitro demonstration of molecular learning with enzymatic weight update. The experiments are designed for plausible pattern recognition with DNA through iterative processes of self-organization, hybridization, and enzymatic weight update through the hypernetwork algorithm. Natural DNA processes act in unison with the proposed molecular learning algorithm using appropriate enzymes which allowed updating of weights to be realized in vitro. Unlike in previous studies, a molecular learning algorithm with enzymatic weight update is proposed, where the positive and negative terms of weight update are considered in the model for learning. Using the validated experimental steps, the model can be used for repeated learning iterations for the selection of relevant DNA to cause the DNA pool to continuously change and optimize, allowing large instance spaces to reveal a mixture of molecules most optimized to function as a DNA pattern recognition classifier.

Our experiments showed a higher order feature extraction method was possible in vitro using higher-order DNA hyperedges which was demonstrated by constructing longer DNA sequence datasets. This method of DNA data construction dramatically reduced the number of unique DNA sequences required to cover the large search space of image feature sets. Finally, DNA ensemble learning is introduced for use of the full feature space in the classification tasks.

Although the complete iterations of learning are yet to be carried out, the aim of this paper was to provide a framework, with a synergistic approach between theoretical and experimental designs of molecular learning algorithm. In future experiments we will carry out the iterative molecular learning scheme wet laboratory conditions.

By harnessing the strength of using biomolecules as building blocks for basic computations, new and exciting concepts of information processing have the potential to be discovered through more molecular computing methods. In turn, the implementation of machine learning algorithms through DNA could also act as a starting point for emerging technologies of computational molecular devices, implicated in a diverse range of fields such as intelligent medical diagnostics or therapeutics, drug delivery, tissue engineering, and assembly of nanodevices. As more advanced applications are explored, more intelligent molecular computing systems, with suitable intelligence to navigate and function in dynamic in vivo environments, may bridge gaps in current molecular computing technologies, so that DNA systems can function in uncontrolled, natural environments containing countless unforeseeable variables.

## Figures and Tables

**Figure 1 molecules-24-01409-f001:**
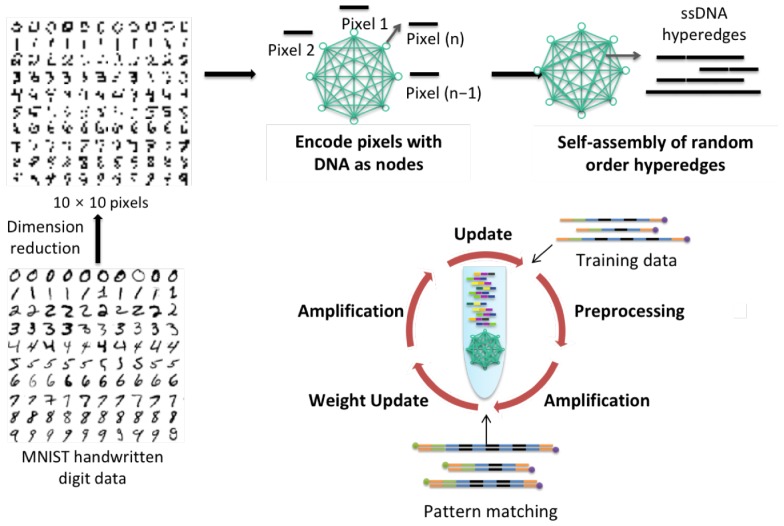
Hypernetwork with nodes and hyperedges with a conceptual overview of molecular machine learning with DNA processes. Each node represents a pixel which is encoded to a unique DNA sequence. These pixel DNA are self-assembled to form random order hyperedges.

**Figure 2 molecules-24-01409-f002:**
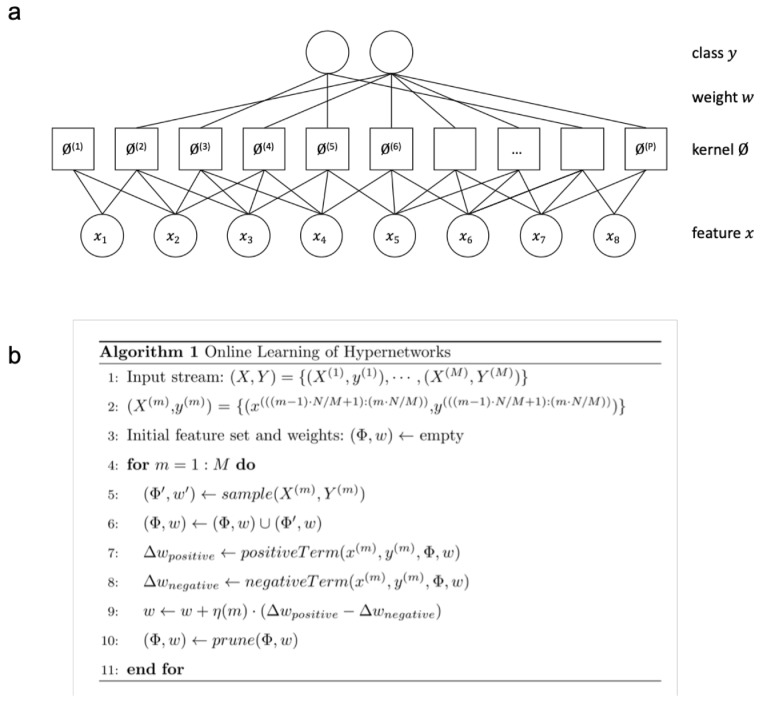
Graphical and algorithmic representation of the hypernetwork model (**a**). The graphical structure of the hypernetwork model. This is the factor graph representation of the model in Figure 1. Note, one kernel corresponds to one factor. (**b**) Algorithm of online learning of hypernetworks.

**Figure 3 molecules-24-01409-f003:**
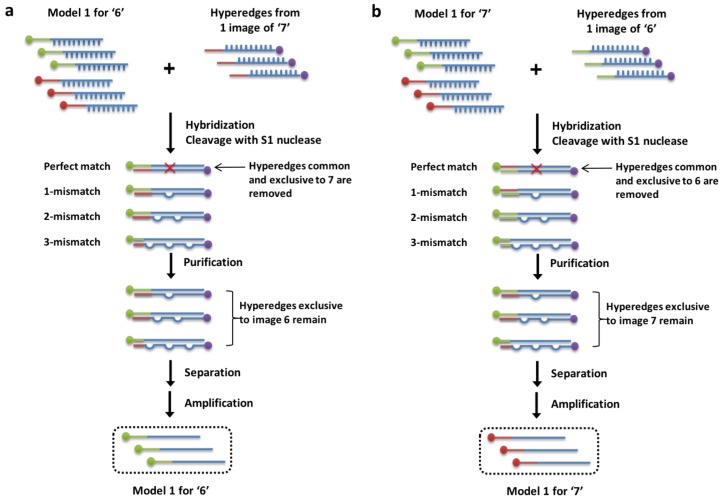
Overview of experimental scheme implementing molecular learning of hypernetworks. (**a**) Experimental steps for training the image ‘6’ (**b**) Experimental steps for training the image ‘7’.

**Figure 4 molecules-24-01409-f004:**
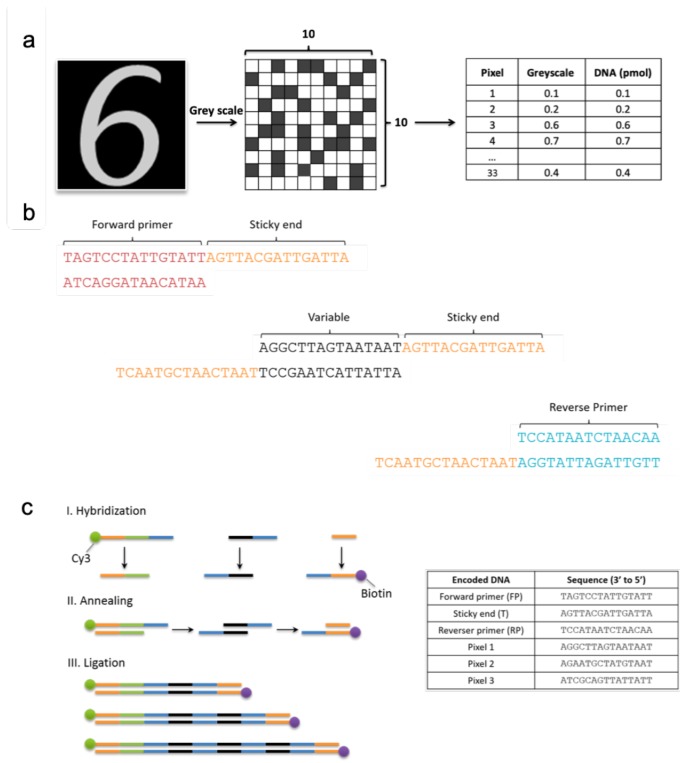
Encoding images to DNA hyperedges (**a**). Encoding of image data to unique DNA sequences (**b**). Complementary DNA sequences of primers, sticky ends and variable DNA that are ligated to form hyperedges (**c**). Self-assembling of free-order hyperedges through three processes, hybridization, annealing, and ligation. Key: Primers (orange), class label (green), tag or sticky end (blue), pixel (black). Please note, the pixel DNA colored black represent unique DNA sequences of various pixels, but for simplicity have been colored black to group them as variable DNA. Table shows DNA sequences for encoded pixels, primers and sticky ends. The final double strands in the sample are separated to single stranded DNA for use as the random DNA library set.

**Figure 5 molecules-24-01409-f005:**
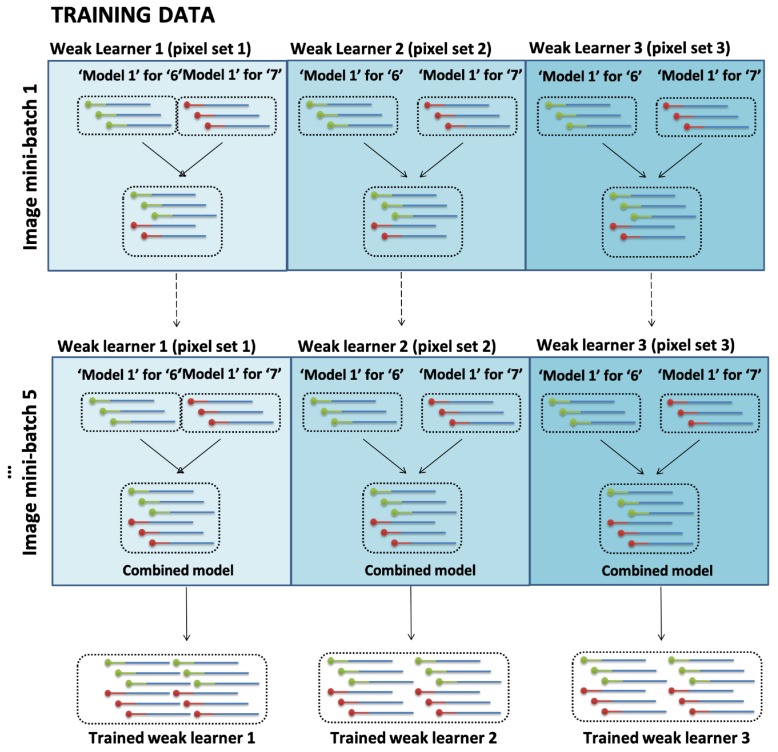
Ensemble of hypernetworks. Three ensembles of both trained models for digit images ‘6’ and ‘7’ are added in an online manner (combined model). The ensembles are added together for ensemble prediction in digit classification in the test stage.

**Figure 6 molecules-24-01409-f006:**
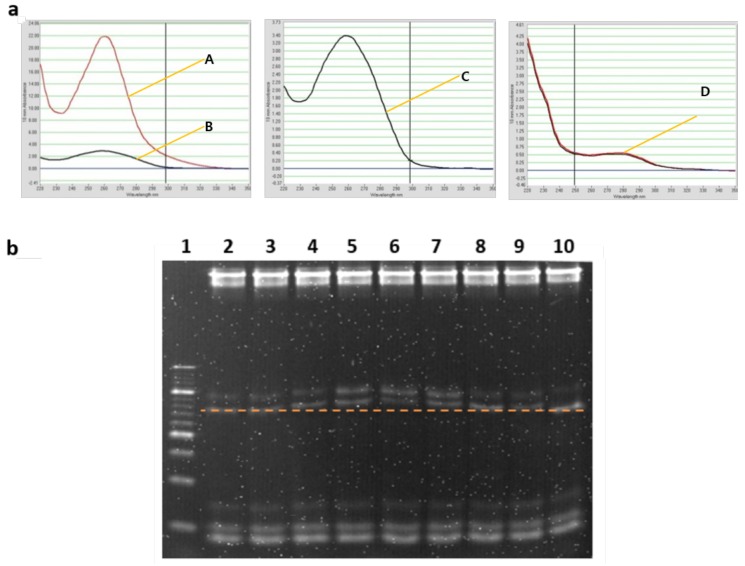
DNA quantification and hybridization of final complementary strands for verifying the presence of a 3-order double-stranded random DNA library. (**a**) Nanodrop analysis for quantification of DNA amount at each step of the protocol; pre-hybridization of variables pre-ligase (A), post-ligase (B), post-purification (C) post-separation (D) (**b**) gel electrophoresis of 3-order random library hybridized with each type of complementary strands (C1-C9 in lanes 2–10), so lane 2 contains the first complementary possible complementary sequence and so on to a total of nine possible hyperedges that could have been produced. Marked line shows 70-mer sequences.

**Figure 7 molecules-24-01409-f007:**
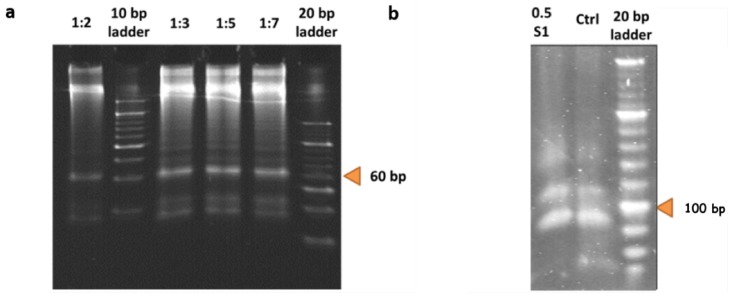
In vitro experimental results (**a**) Free-order hyperedge production. The 10 bp and 20 bp ladders were added in lanes 2 and 6. The ratio between primers and variable DNA sequences were varied to see if there was any effect on the length of the hyperedges produced. (**b**) The control lane shows from 70 base pairs up, perfect, 1, 2, 3-mismatched sequences. The perfectly matched DNA at 70 bp, 1-mismatch at 90 bp, 2-mismatched at 110 bp and 3-mismatched at 130 bp. Lane 1 contains the sample with S1 nuclease treatment. The perfectly matched DNA strands in the lower most band was cleaved.

**Figure 8 molecules-24-01409-f008:**
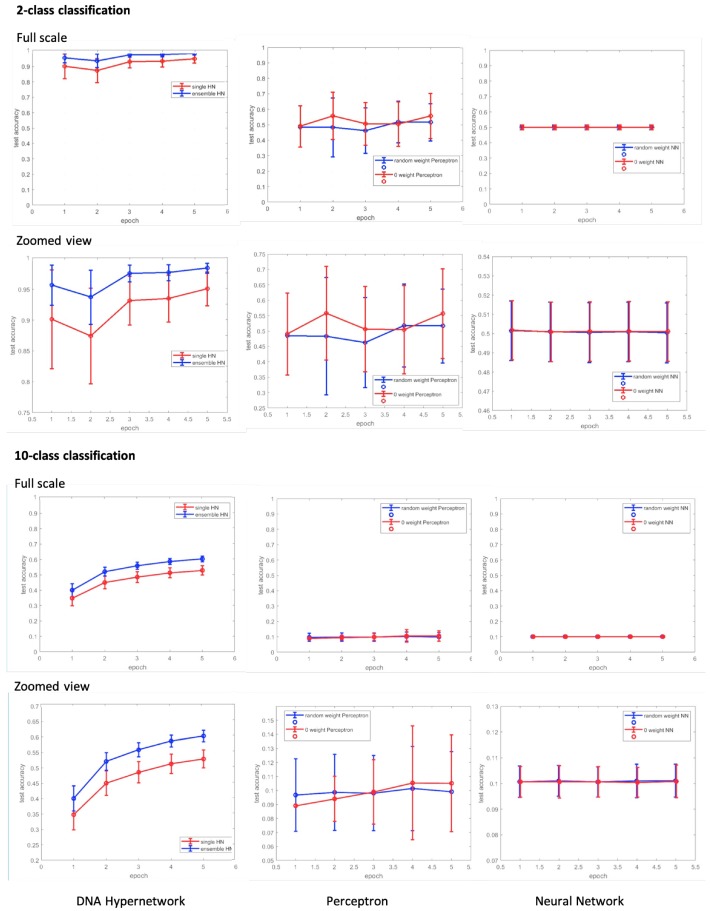
In silico experimental results. Computer simulation of 2-class classification with hypernetwork d. Computer simulation of 10-class classification with hypernetwork.

**Figure 9 molecules-24-01409-f009:**
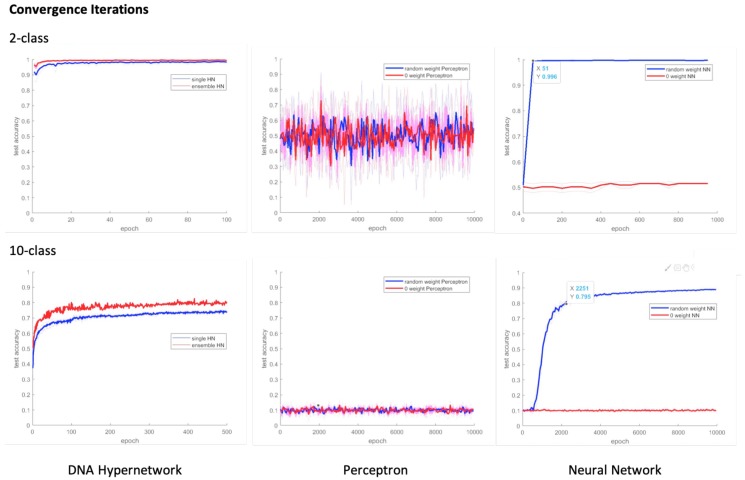
In silico experimental results. Computer simulation of 10-class classification with hypernetwork.

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
