# Peer review of "Enzymatic Weight Update Algorithm for DNA-Based Molecular Learning"

_molecules, 2019, doi:10.3390/molecules24071409_

Round 1

Reviewer 1 Report

The paper is about using DNA computations to correct a machine learning algorithm that will be used on a standardized hand-written data sheet of the number 1-9 that was moved into a 10 x 10 pixel format. Numbers 6 and 7 were used as the experimental focal points. The machine learning algorithm will select a random number for number 6 and create a 10 x 10 pixel format and map it to a DNA strand to be used in the algorithm for processing.

A weakness I noticed about the paper is it mainly focused on actual implementation of the machine learning algorithm without enough background presentation. Although this is not a pure bad thing I had a lot of trouble following it because it did not explain clearly what some of the data were trying to display. For example Figure 5 and Figure 6.

                The figures used in the paper to show the DNA computations were easy to follow but using the same color for parts of the strand was confusing. This makes me interpret that part of the DNA strands for the number 6 and 7 are the exact same although this could be a false statement I deducted.

Overall, the paper seems to be on the right track. It displays enough information about the machine learning language to have an idea about what’s going it. It displays actual implementation results. I recommend the authors add some recent similar papers in performing machine learning with DNA and compare them with their work. The following paper is one good example :

Computing mathematical functions using DNA via fractional coding

SA Salehi, X Liu, MD Riedel, KK Parhi

Nature publication, Scientific reports 8 (1), 8312-(2018)

Author Response

Reviewer 1

The comments put forward by the reviewer in response to our paper were greatly appreciated and have contributed greatly to the improvement of our work. We have read and accepted the concerns put forward and have detailed below our responses.

In response to the reviewer's comment that the experimental results section (Figure 5 and Figure 6) are lacking detail, and was unclear, we recognize this as a valid concern. To remedy this, we have expanded on the experimental results sections to supplement and support the detail existing in the original submission. This will enable replication of the experiments conducted, confirming the validity and quality of our results.

-       We have replaced Figures 7 c and d with improved simulation results and written a more in-depth description of the simulation details. This can be found in Section 3.4. Performance of in silico experiment of DNA hypernetwork (Lines 293-341). The source code has also been uploaded on GitHub (https://github.com/drafity/dnaHN) and referenced to in the text (line 336)

Summary of all edits made:

-       The next step of the algorithm consists of pattern matching and weight update (Algorithm 1, line 7, 8). Equation 3 shows the gradient of the log-likelihood of Equation 1 (line 224-225)

-       Related works of (Cherry and Qian, Nature, 2018) (line 44-52)

-       Reference to figure 2 (line 124)

-       Further description about gradient descent (line 132-134)

-       Explanation about Algorithm 1 in relation to the experimental scheme in Figure 4 (line 142-148)

-       Addition of figure 2 regarding our hypernetwork model (page 5)

-       Description of Algorithm 1 and Equation 3 (line 221-222)

-       Figures 7 c and d with the new simulation results comparing our work to a neural network and the perceptron used in the work of (Cherry and Qian, Nature, 2018)

-       Improved Section 3.4 with updated in silico learning results compared to other learning models (Line 317-367)

-       Link to the uploaded source code on GitHub repository (line 336)

Figure 2 has also been updated to clarify that the same color parts of strands indeed refer to unique DNA strands, but for simplicity have been colored black to discriminate it as the variable DNA strand as opposed to linking DNA or primer DNA (please refer to Figure 2 legend).

Finally, we have added a short comparison of our work to the paper suggested by the reviewer (SA Salehi, et al.) in the introduction section as we deemed this a similar study with shared values in implementing machine learning with DNA (line 74-80)

The authors would once again like to express our gratitude to the reviewer as we can confirm that the edits made in response to the reviewer’s comments have improved our paper.

Reviewer 2 Report

In this paper, the authors build upon their previous work to implement a machine-learning algorithm in a molecular programming framework. Their framework is a DNA-based implementation of hypernetworks. Computation is carried out by hybridizing single-stranded input DNA strands to the hyperedges of the network. A completely double-stranded hyperedge is considered to be an output signal of the system. 

The algorithm they describe here works by using a nuclease to remove false positives. Then, they use a polymerase chain reaction (PCR) to amplify the remaining DNA strands. The nuclease and PCR thus implement the weight update of the network, which can then be used for the next round of training.

They apply this approach to show a theoretical way to train a hypernetwork to distinguish between the '6' and '7' digit from the MNIST library. After providing some preliminary in-vitro results, they use a simulation of the system to show that it should behave correctly. They provide similar results (with a lower performance accuracy, however) for the full 10 digits set. While no detail is provided, the authors mention that they simply scaled their approach.

I have two main concerns about the manuscript:

* Some key references are missing. In particular, the omission of Cherry and Qian, Nature, 2018 is glaring, as their work has a very similar goal (classifying digits from the MNIST library) and similar methods. In fact, those similarities go all the way down to the specifics: discretizing the digits on a 10x10 grid, using only 20 pixels to limit the search space, focusing first on the distinction between the '6' and '7' digits, and so on. While I do believe that the authors are writing in good faith, an extensive discussion of the difference in approach and results is necessary. One axis to follow could be that Cherry and Qian chose to perform the learning in-silico, while the authors are aiming for in-vitro.

* A second concern, however, is that the authors did not, in fact, perform in-vitro learning. While they did perform some promising proof-of-concept validation, the only learning results come from a simulation of the system (Section 3.4). Moreover, the details of the simulation were left out from the text, making it impossible to judge its pertinence or accuracy. My recommendation would be, if possible, to actually perform the experiments, and show that they behave similarly to the simulation. At the very least, the authors should describe the in-silico model in more details, and, ideally, provide a link to source code.

Minor concerns:

* It is surprising, considering the nature of their computing framework, that the authors do not cite Adleman, L. M. (1994). Molecular computation of solutions to combinatorial problems. Science, 266(5187), 1021-1024.

* The authors only focus on rule-based and logic gate-based systems and do not mention enzyme-based computing paradigms implementing reaction networks, such as:

** Genelets (e.g. Kim, J., & Winfree, E. (2011). Synthetic in vitro transcriptional oscillators. Molecular systems biology, 7(1), 465.)

** DNA PEN Toolbox (e.g. Montagne, K., Plasson, R., Sakai, Y., Fujii, T., & Rondelez, Y. (2011). Programming an in vitro DNA oscillator using a molecular networking strategy. Molecular systems biology, 7(1), 466.)

** EXPAR (e.g. Van Ness J,  Van Ness LK,  Galas DJ. Isothermal reactions for the amplification of oligonucleotides, Proc. Natl Acad. Sci. USA, 2003, vol. 100 (pg. 4504-4509), Qian, J., Ferguson, T. M., Shinde, D. N., Ramírez-Borrero, A. J., Hintze, A., Adami, C., & Niemz, A. (2012). Sequence dependence of isothermal DNA amplification via EXPAR. Nucleic acids research, 40(11))

All of those are fit for implementing dynamic reaction networks, which are similar in concept to the authors' hypernetworks, and thus could be good candidates to implement machine learning algorithms.

* The authors should prove that their system correctly implements (at least an approximation of) Equation 1. They should also show that their algorithm does perform updates according to Equation 3. Lines 180 to 185 only say that said equation would be appropriate for updating the hypernetwork. Specifically, while the nuclease and PCR are indeed providing a negative and positive weight update, respectively, nothing in the text proves that the update follows Equation 3.

* The flow of the article is a bit strange. Figure 4 is mentioned before Figures 2b, 2c and Figure 3. Meanwhile, the reader has to go back to Figure 1 later on to have an overview of the algorithm.

* There are more than a few English errors, and the style can be difficult to understand at times. I would recommend using professional proof-reading before the final submission.

To conclude, I think the work is of interest, but cannot be published without major revisions.

Author Response

Reviewer 2

The authors would like to sincerely extend their gratitude to the reviewer for their constructive comments and the detail contained within them. The reviewer’s suggestions have substantially contributed to the improved quality of our paper. In response to the comments put forward, we have tried our best to address the two main concerns of the reviewer, the inclusion of key reference (Cherry and Qian, Nature, 2018), and more in-depth explanation of our in silico learning results.

We have discussed and performed new in silico experiments comparing our model to the model presented in (Cherry and Qian, Nature, 2018) and a neural network in general. We agree that this is a key reference which should have been included and further investigated in relation to our work. As background information, at the time we were performing our experiments and preparing our paper, Cherry’s paper had not been published yet and although some details were overlapping, we thought our model and goals of our study to implement in vitro learning had key differences. However, upon reading the reviewers comments it seemed crucial to compare our work with this state-of-the-art study of DNA recognition of Cherry’s paper.

We have replaced Figures 7 c and d with the new simulation results comparing our work to a neural network and the perceptron used in the work of Cherry and Qian. We have written a more in-depth description of the simulation details as suggested as a second main concern by the reviewer. This can be found in Section 3.4. Performance of in silico experiment of DNA hypernetwork (lines 293-341). The source code has also been uploaded on GitHub (https://github.com/drafity/dnaHN) and referenced to in the text (line 336).

Summary of all edits made:

-       The next step of the algorithm consists of pattern matching and weight update (Algorithm 1, line 7, 8). Equation 3 shows the gradient of the log-likelihood of Equation 1 (line 224-225)

-       Related works of (Cherry and Qian, Nature, 2018) (line 44-52)

-       Reference to figure 2 (line 124)

-       Further description about gradient descent (line 132-134)

-       Explanation about Algorithm 1 in relation to the experimental scheme in Figure 4 (line 142-148)

-       Addition of figure 2 regarding our hypernetwork model (page 5)

-       Description of Algorithm 1 and Equation 3 (line 221-222)

-       Figures 7 c and d with the new simulation results comparing our work to a neural network and the perceptron used in the work of (Cherry and Qian, Nature, 2018)

-       Improved Section 3.4 with updated in silico learning results compared to other learning models (Line 317-367)

-       Link to the uploaded source code on GitHub repository (line 336)

-       Reordering of figures for a more natural flow

As a final note, all wet-lab molecular learning experiments have been carried out, complete to show significant accuracy in classification. However, as much as we wish to include this in this paper, verification results from the DNA sequencing team have been held up due to technical reasons for some time and we have decided to publish this in another separate paper in the future.

Nonetheless, thank you to your constructive feedback, we are now happy to present to you, this paper which introduces a novel molecular learning algorithm suitable for in vitro implementation, validates the experimental protocol through wet lab experiments and demonstrates competitive classification results to current leading DNA learning models such as the neural network and the winner-take-all model.

Additional minor concerns have also been addressed:

-       Inclusion of Adelman’s paper (line 21 – 22)

-       Addition of dynamic reaction networks references and discussion in relation to our work (lines 53-66)

-       To bridge the gap between theoretical and practical protocols, we added an explanation about Algorithm 1 in relation to the experimental scheme in Figure 4 (line 142-148)

We have gone through the paper again for English errors, but as suggested by the reviewer, if accepted we will definitely use professional proof-reading before the final submission.

The authors would once again like to thank the reviewer as we believe that the edits made in response to the reviewer’s comments has helped significantly in improving our paper.

Reviewer 3 Report

It was my great pleasure to read the paper entitled "Enzymatic Weight Update Algorithm for DNA-Based Molecular Learning". The authors successfully address an interesting and promising topic within the field of in vitro molecular computing by convincing introduction of a novel approach towards a wetlab implementation of machine learning based on DNA strands. Its main idea consists in an algorithmic scheme in which a neural network including its weights can be entrained using techniques hybridisation, ligation, and cleavage. Along with conduction of the learning process, the weights of the artificial neurons get adjusted by successive modification of the DNA pool. The amount and the structure of DNA strands in the underlying pool encode the weights. PAGE gel images demonstrate the practicability of the approach. Although the main ideas have been convincingly described and illustrated, in my opinion the paper lacks an explanation for reconstruction of the finally entrained neural network topology and its weights from the final PAGE gel image. This step seems to be important for general employment of the approach since it provides new information and makes the neural network applicable. Perhaps, the PAGE gel image needs to be complemented by additional analysis techniques in order to do the reconstruction in detail. Altogether, the paper is well-written and an asset to the community of molecular computing. Unequivocally, it is worth to be published after minor revision.

Typos:

p. 5, caption of Figure 5, l. 6: single straded DNA -> single stranded DNA

p. 7, caption of Figure 4, l. 1: both trained model -> both trained models

p. 12, l. 201: for the selection for the selection -> for the selection

o. 12, l. 206: datsets -> datasets

Author Response

Reviewer 3

Thank you for your positive review of our paper. It has lifted our spirit to further improve our work. The comments put forward by the reviewer in response to our paper were greatly appreciated and have contributed greatly to the improvement of our work. We have read and accepted the suggestions put forward and have detailed below our responses.

The reviewer has expressed a lack of explanation of reconstruction of the finally entrained neural network topology and its weights from the final PAGE gel image. The final PAGE gel image represents the workings of S1 nuclease to cleave mismatched DNA, which plays a key role in implementing molecular weight update.

The reasons for which DNA strands with different number of mismatched bases are, that although the molecular weight is the same, the bubbles cause a structural hindrance in the gels and thus causes a difference in run time. Additional references were added to describe this phenomenon.

-       Zacharias, M., Hagerman, P. J., 1996. The influence of symmetric internal loops on the flexibility of RNA. Journal of molecular biology 257 (2), 276–289.

-       Zeng, Y., Zocchi, G., 2006. Mismatches and bubbles in DNA. Biophysical Journal 90 (12), 4522–4529.

This gel image (Figure 7 b) does not contain the finally entrained neural network topology, but is a proof of the workings of the S1 nuclease. The finally entrained hypernetwork topology is demonstrated in Figure 2 and Figure 7 c and d in the classification accuracy results. We have added a much more descriptive section under 3.2 to discuss the details of the experiments carried out in hopes that this clarifies our work (lines 293-341). The source code has also been uploaded on GitHub (https://github.com/drafity/dnaHN) and referenced to in the text It is worth mentioning that the reconstruction of the hypernetwork is possible with DNA sequencing results from the trained DNA model. Here, we focus on introducing our molecular learning model and provide proof-of-concept experiments. However, as the reviewer has described, the DNA hyperedges and weights of these hyperedges can be derived through next generation sequencing of DNA samples, and thus used to reconstruct the hypernetwork.

Summary of all edits made:

-       The next step of the algorithm consists of pattern matching and weight update (Algorithm 1, line 7, 8). Equation 3 shows the gradient of the log-likelihood of Equation 1 (line 224-225)

-       Related works of (Cherry and Qian, Nature, 2018) (line 44-52)

-       Reference to figure 2 (line 124)

-       Further description about gradient descent (line 132-134)

-       Explanation about Algorithm 1 in relation to the experimental scheme in Figure 4 (line 142-148)

-       Addition of figure 2 regarding our hypernetwork model (page 5)

-       Description of Algorithm 1 and Equation 3 (line 221-222)

-       Figures 7 c and d with the new simulation results comparing our work to a neural network and the perceptron used in the work of (Cherry and Qian, Nature, 2018)

-       Improved Section 3.4 with updated in silico learning results compared to other learning models (line 317-367)

Thank you for mentioning the typos, all have been corrected accordingly.

The authors would once again like to thank the reviewer as we believe that the edits made in response to the reviewer’s comments has helped significantly in improving our paper. 

Round 2

Reviewer 1 Report

The authors tried to address my concerns. However, I still think the introduction part can be improved by adding a more inclusive and detailed narrative of prior work and deeper comparison with this work.

Minor concerns:

If it's possible, make figure 5 more informative. Otherwise, I suggest to compact it since, except their background color's intensity, the color boxes in the figure are identical. 

English language and style are fine/minor spell check required particularly for the added parts in the introduction.

Author Response

Reviewer 1

The authors would like to sincerely extend their gratitude to the reviewer for their constructive comments. The reviewer’s suggestions have substantially contributed to the improved quality of our paper. In response to the comments put forward, we have tried our best to address the two main concerns of the reviewer, the inclusion of more references of prior work and comparison with this work, and reconfiguration of figure 5.

Please see that the introduction section has been revised from Line 54-87 to include more work and relevance to our work. As we were pressed for time, please understand that some of the responses are direct quotations of the refined sections of the paper.

Lines 54-87:

“Another related area of research includes the implementation of dynamic reaction networks. \cite{kim2011synthetic} use \textit{in vitro} biochemical systems, transcriptional circuits to form complex networks by modifying the regulatory and coding sequence domains of DNA templates. A combination of switches with inhibitory and excitatory regulation is used to oscillators similar to that which is found as natural oscillators. \cite{montagne2011programming} also use chemical reactions inspired from living organisms to demonstrate assembling of a \textit{de novo} chemical oscillator, where the wiring of the corresponding network is encoded in a sequence of DNA templates. These studies use the synthetic systems to further understand the complex chemical reactions found in nature to deepen our understanding of the principle of biological dynamics. A key similarity to our work is the use of modular circuits to model more complex networks. However, it is important to note that these studies are all demonstrated \textit{in silico}, although it illustrates the potential of \textit{in vitro} transcriptional circuitry. Computational tools are also being developed, one example being the EXPonential Amplification Reaction (EXPAR), to facilitate the assay design of isothermal nucleic acid amplification \cite{qian2012sequence}. This method helps accelerate DNA assay design, identifying template performance links to specific sequence motifs.

These dynamic system programming paradigms could be valid approaches to implement machine learning algorithms, as programmable chemical synthesis and the instruction strands of DNA dictate which reaction sequence to perform. We ponder that this kind of powerful information-based DNA system technology could also be manipulated to perform defined reactions in specific orders similar to what our study strives to do, thus, implementation operations \textit{in vitro} to demonstrate molecular learning with the hypernetwork or other machine learning algorithms \cite{fu2018dna}.

Recent work by \cite{salehi2018computing}, implement mathematical functions using DNA strand displacement reactions. This study demonstrates considerably more complex mathematical functions to date, can be designed through chemical reaction networks in a systematic manner. It is similar to our work in that it strives to compute complex functions using DNA though a key difference is that the design and validation of this work were presented \textit{in silico} whereas our work focuses on \textit{in vitro} implementation of molecular learning. However, the mass-action simulations of the chemical kinetics of DNA strand displacement reactions may be key in developing \textit{in vitro} learning implementations, as learning consists of target mathematical operations which need to be performed with DNA in a systematic manner to manipulate DNA datasets. Consequently, operations or computational constructs are crucial in implementing machine learning algorithms, from simple perceptrons to neural networks and this is proposed by this system, thus shares our interests in building systemic molecular implementations of chemical reactions for molecular machine learning. Further examples include a study where an architecture of three autocatalytic amplifiers interacts together to perform computations \cite{song2017design}. The square root, natural logarithm and exponential functions for x in tunable ranges are computed with DNA circuits.”

As advised, we have made figure 5 more compact to reduce the redundancy of the iterations of training.

Also, we have significantly revised Section 3.4 and added figures 8 and 9 following advice from the other reviewer. We hope that the added results and adjoining detailed discussion section can help you reconsider if the methods are adequately described and if the conclusions are supported by the results!

We apologize for the small errors in our writing, we were pressed for time before the last submission and again for this resubmission in fact, but have made sure to thoroughly proofread the writing of the revised version.

Thank you once again for your valued advice as we believe it has improved our paper, especially the introduction section, following the last resubmission. We hope you agree and that most of your concerns have been addressed for a better outcome for our paper.

Thank you for your time and consideration of our team.

Reviewer 2 Report

While the authors tried to address comments, I do not find their edits satisfying. I assume that the issue lies more in the timeframe allowed for the resubmission rather than in the authors' willingness to improve the paper. Therefore, I would suggest to the editor leaving more time for further edition (but will defer to their decision in any case).

With respect to my major concerns:

* The description of the work of Cherry and Qian is very shallow. The authors make no mention of the performance achieved by their approach. The authors should also justify better why their approach is performing substantially worse. If online learning on a small number of epochs is the main reason, the authors could also mention an empirical optimal performance of their hypernetwork by simulating a hundred or even a thousand epochs. 

* The authors did add their algorithm and tried to relate it to the molecular implementation in Figure 3. While I am mostly satisfied with their corrected version, the description and algorithm do not seem to match perfectly. The description of Figure 3 seems to indicate that weight updates are performed sequentially for both targets, which is corroborated by the description of the algorithm in the text. However, the algorithm itself has a single weight update. It is also unclear how the pruning part would be performed in-vitro. The text mentions eliminating hyperedges with negative weights. Does that imply that they would have a (theoretically) negative concentration, thus get pruned "for free"?

I have a new major concern from the authors' negative controls in Figure 7. First of all, it is extremely strange that all models are performing no better than a random guess (1/2 correct for 2 classes, 1/10 correct for 10 classes). Learning also seems suspiciously inefficient. As a side comment, setting all initial weights to 0 is usually considered bad practice, especially without symmetry breaking elements. The authors also mentioned performing a thousand replicates of each experiment. In that case, I would also recommend adding error bars to the Figure to get a better idea of how noisy the learning process can be. Finally, I would again suggest to dramatically increase the number of epochs, to show how long it would take other systems to take over the authors' approach. 

With respect to minor concerns:

* While the authors did include references to the dynamic system programming paradigms I listed, they only gave a very general description of each (lines 53-66). The authors do not have to mention explicitly the results from those papers, but should instead mention that those would be valid approaches to implement hypernetworks (or the authors' machine learning algorithm) as well.

* The authors did not address the following concern:

"The authors should prove that their system correctly implements (at least an approximation of) Equation 1. They should also show that their algorithm does perform updates according to Equation 3. Lines 180 to 185 only say that said equation would be appropriate for updating the hypernetwork. Specifically, while the nuclease and PCR are indeed providing a negative and positive weight update, respectively, nothing in the text proves that the update follows Equation 3."

They should explain how the different elements of the equations relate to biochemical reaction rates (in particular enzymatic rates).

Comment:

* I could confirm that the authors' code is available online, however, readme section was lacking. There is no explanation of the different scripts, nor is there a list of the required Matlab toolboxes. I could not run the code as I seem to be missing one of them.

To conclude, I would like to reiterate that I believe this work to be of interest to the readership of Molecules. While the authors seem on the right track, I do think that some additional work is still required before publication.

Author Response

Reviewer 2

The authors would like to thank the reviewer for the detailed comments and advice once again. We valued each and every one of your comments and have tried our best to address the issues in the given time, which we believe has helped us significantly improve our paper. As we were pressed for time, please understand that some of our responses are direct quotations of the refined sections of the paper.

Firstly, we have focused on adding a more comprehensive description of the work of Cherry and Qian as advised, and have incorporated better justifications when comparing our work to theirs, in terms of differences in the performance. To do this, we have added improved in silico experimental data to more thoroughly demonstrate when and why our model performs better than the perceptron and the neural network in the smaller number of epochs, and better than the perceptron but not the neural network after a large number of epochs. As advised, we dramatically increased the number of epochs, presenting convergence of all models and have added error bars to all data to illustrate the noise present.

Line 52-53

“This is further discussed in the results section as a comparative study with our work (Section 3.4).”

Line 349-365

“In \cite{cherry2018scaling}, a basic perceptron model outputs the weighted sum for each class and selects the maximum value as their winning final output. 2-class classification between digits ‘6’ and ‘7’ is demonstrated and 9 label 3 grouped class-classifier is described, where all methods first eliminate the outlier and the performance achieved by providing probabilistically calculated weights of the 10 most characteristic features to the designer as a prerequisite. However, in our study, we do not eliminate outliers or give prior weights and use the MNIST dataset as it is for our performance. We exploit the learning ability of a DNA computing model without the need for the designer to previously define weights. Not only do we reduce the labor required by the designer to define weights for selected features but we exploit the massively parallel processing capability of DNA computing whilst demonstrating molecular learning which improves performance with experience as our model is designed for implementation in vitro through molecular biology technique with wet DNA.

Two types of initialization of weights are introduced in our simulation results, 0 weight initialization which is easily implemented in DNA experiments and random weight initialization which is harder to be conducted in vitro but is more conventionally used in perceptron and neural network models. The perceptron and neural networks convergence of performance are dependent on their initialized weights \cite{nguyen1990improving}. We conducted these two methods of initializing the weights, first, starting with 0 weight and second, providing random values to the weights.”

Line 419-422

“We also discuss the reasons why the perceptron does not perform as well. Due to the nature of perceptron models, as a representative linear classifier, it is difficult for it to solve linearly inseparable problems (XOR problems) without any preprocessing or adding layers to the perceptron to deal with non-linearity problems \cite{rosenblatt1957perceptron, russell2016artificial}”

Line 423-440

“As illustrated in Figure 8, the perceptron model shows performances close to what would be achieved from random picking for both small and large number of iterations. However, depending on how the data is fed, 2-class classification performance levels show major fluctuations, where up to 80% performance is achieved at times and others much lower performance. This phenomenon is typically representative of unsuccessful generalization of the data also called overfitting. For example, in the case of the perceptron, as described in the reference, the performance is achieved only for the data that can be fitted linearly. To learn linearly inseparable data, the model needs a feature reduction or extraction preprocessing methods \cite{liu2003handwritten} or a nonlinear kernel to model (e.g. Support Vector Machine \cite{scholkopf2001learning}, Neural Network \cite{russell2016artificial} the high dimension input dataset. As this paper focuses on the implementation of a learning model in vitro only using easy, basic and fundamental learning processes, we believe this is out of the scope of our paper and omit further discussion.

As a support to such arguments, as shown in figures 8 and 9, both in our results and in Cherry and Qian’s work, there are cases where a variety of elimination conditions and previously providing the optimal weights of batch data by the designer can achieve significant performance in 2-class classification i.e. overfitted (the maximum value of the error bars). However, as in the case of 10-class classification tasks, where the data is not linearly separable where it exceeds the model's capacity, the range of performance levels are smaller and, as acknowledged by Cherry and Qian in their paper, it is difficult for the designer to find the optimal weights for the model.”

To address the concern regarding the models performing no better than a random guess, we explain that models can perform worse than a random guess when adjusting to noise, where it is valuing the suboptimal/incorrect features. This can occur in cases where fundamental assumptions are violated or there is an imbalance when using accuracy as the baseline. However, in our data, the fluctuations in performance, where in some cases the model performs no better than a random guess can be explained through overfitting in the perceptron model as explained above. Again, this is why feature extraction, feature reduction or the biasing of weights to certain values are developed in machine learning, used to preprocess the data for better performance. However, as we are pursuing to implement molecular machine learning in vitro, which is as scalable and autonomous as possible, introducing bias or different forms of preprocessing was deemed ineffective in our design.

Furthermore, as we define the weight of hyperedges by the concentration of that DNA in the test tube, it is not possible for a DNA hyperedges to have a negative weight in our model. Hence, initializing the weights to 0 was an easier and more appropriate method for the implementation of the molecular hypernetwork and we compare the in silico simulation results with other existing methods.

Line 406-418

“The hypernetwork, inspired by DNA chemical reactions, when computed in silico, clearly showed the disadvantage of sequential computing in silico and the massively parallel processing advantage of DNA computing in vitro. In an instant, DNA molecules hybridize when complementary strands are added together in an appropriate buffer and thus almost immediately the computing in that tube comes to an end. However, implementation of the hypernetwork in silico, is iterative, sequential. For each training and test data, the number of matches and mismatches need to be calculated sequentially, and when the number and the order of hyperedges increases, computational time complexity increase exponentially. As a result, with our computing power, empirically, 1000 iterations require 1000*20 minutes, a total of approximately 14 days to complete. Therefore, it is important to note that there is a sheer advantage in DNA implementation of the hypernetwork compared to in silico. For the same reason, the neural network requires around 10000 iterations to converge and in the case of non-linearly separable data when using the perceptron model, it fails to converge. Thus, the proposed hypernetwork may also be introducing the possibility of a new computing method.”

The next concern was regarding Figure 3 and confusion in how it links to Algorithm 1. You are correct in that Figure 3 illustrates that the negative and positive weight update sequentially. The negative weight update occurs with the S1 nuclease cleavage and positive weight update in the subsequent amplification of remaining DNA hyperedges (both top and bottom arrows of both Figure 3a and 3b).

It is also true that line 9 of Algorithm 1, weight update is presented in one line as a single weight update. This is because in lines 7 and 8, the positive and negative terms are calculated for each hyperedge and then weights updated.

It is understandable that confusion could have been aroused and we have added additional explanations to clarify that there are sequential differences in the in silico and in vitro algorithm. We have also tried to accentuate that both systems as a whole implement the same idea of negative and positive weight update, and that this is the novelty in our work. We have highlighted the significance of designing a novel in vitro algorithm and complementary practical molecular scheme which contain the positive and negative term calculations for update, conventionally practiced in machine learning.

Line 158

1. Initializing hyperedge in each epoch corresponds to line 4

2. Figure 3 hybridization corresponds to line 7 - 8

3. Figure 3 nuclease and amplification corresponds to line 9

4. Merging hyperedges in each epoch corresponds to line 10

Line 162-168

“In the in vitro implementation of Algorithm 1, the updating of calculated positive or negative term occurs in a slightly different order. In the case of negative weight update, the nuclease cleaves the perfectly matched DNA strands, which occur from the hybridization of complementary DNA hyperedges from training data for ‘6’ and ‘7’, the hybridization being when the negative weight term is calculated. In the case of the positive weight update, the resulting DNA concentrations of each hyperedge from cleavage and purification is amplified, where the positive term was also calculated from the initial hybridization process.”

The pruning process of the molecular experiment takes place with the S1 nuclease enzyme weight update operation, where the nuclease cleaves the best matches sequences from the hybridized pool of hyperedges from the DNA pool of digits ‘6’ and ‘7’. The hyperedges that are best matched and cleaved is representative of those which are common to both classes and thus not explicit to either class for use in classification.

In our experiments, the weights are always 0 or above as there is no negative DNA concentration to represent negative weights. It is the negative weight update process which leads to the cleavage of select hyperedges and subsequent decrease in concentration of those hyperedges which represents the decrease in weight of the hyperedges.

In the many variations of the hypernetwork \cite{ha2015automated, lee2013evolutionary, ha2012text}, there are methods of giving negative weights, but as this paper focuses on the implementation the hypernetwork in vitro through DNA, the in silico model also uses a base weight of 0. This is also one of the reasons why we compare the 0 weight models in our results section. Most models encounter issues (convergence to local optima) when given 0 weights, however, we propose a method of addressing this issue with the DNA hypernetwork.

With respect to the minor concern regarding the added references on the dynamic system programming paradigms, we have included a few more lines comparing these works to our goals.

Lines 54-87:

“Another related area of research includes the implementation of dynamic reaction networks. \cite{kim2011synthetic} use \textit{in vitro} biochemical systems, transcriptional circuits to form complex networks by modifying the regulatory and coding sequence domains of DNA templates. A combination of switches with inhibitory and excitatory regulation is used to oscillators similar to that which is found as natural oscillators. \cite{montagne2011programming} also use chemical reactions inspired from living organisms to demonstrate assembling of a \textit{de novo} chemical oscillator, where the wiring of the corresponding network is encoded in a sequence of DNA templates. These studies use the synthetic systems to further understand the complex chemical reactions found in nature to deepen our understanding of the principle of biological dynamics. A key similarity to our work is the use of modular circuits to model more complex networks. However, it is important to note that these studies are all demonstrated \textit{in silico}, although it illustrates the potential of \textit{in vitro} transcriptional circuitry. Computational tools are also being developed, one example being the EXPonential Amplification Reaction (EXPAR), to facilitate the assay design of isothermal nucleic acid amplification \cite{qian2012sequence}. This method helps accelerate DNA assay design, identifying template performance links to specific sequence motifs.

These dynamic system programming paradigms could be valid approaches to implement machine learning algorithms, as programmable chemical synthesis and the instruction strands of DNA dictate which reaction sequence to perform. We ponder that this kind of powerful information-based DNA system technology could also be manipulated to perform defined reactions in specific orders similar to what our study strives to do, thus, implementation operations \textit{in vitro} to demonstrate molecular learning with the hypernetwork or other machine learning algorithms \cite{fu2018dna}.

Recent work by \cite{salehi2018computing}, implement mathematical functions using DNA strand displacement reactions. This study demonstrates considerably more complex mathematical functions to date, can be designed through chemical reaction networks in a systematic manner. It is similar to our work in that it strives to compute complex functions using DNA though a key difference is that the design and validation of this work were presented \textit{in silico} whereas our work focuses on \textit{in vitro} implementation of molecular learning. However, the mass-action simulations of the chemical kinetics of DNA strand displacement reactions may be key in developing \textit{in vitro} learning implementations, as learning consists of target mathematical operations which need to be performed with DNA in a systematic manner to manipulate DNA datasets. Consequently, operations or computational constructs are crucial in implementing machine learning algorithms, from simple perceptrons to neural networks and this is proposed by this system, thus shares our interests in building systemic molecular implementations of chemical reactions for molecular machine learning. Further examples include a study where an architecture of three autocatalytic amplifiers interacts together to perform computations \cite{song2017design}. The square root, natural logarithm and exponential functions for x in tunable ranges are computed with DNA circuits.”

Finally, the minor concern regarding the lack of proof that our system follows Equations 1 and 3 came up again. We thought we had addressed this with the other revisions made in revision 1, however, following your comment in review 2, we have thought hard about how we could prove that our system correctly implements or approximates Equation 1 and 3.

First, the link between the algorithm and our molecular experimental scheme is reiterated:

Line 145

1. Initializing hyperedge in each epoch corresponds to line 4

2. Figure 3 hybridization corresponds to line 7 - 8

3. Figure 3 nuclease and amplification corresponds to line 9

4. Merging hyperedges in each epoch corresponds to line 10

Line 149-155

“In the in vitro implementation of Algorithm 1, the updating of calculated positive or negative term occurs in a slightly different order. In the case of negative weight update, the nuclease cleaves the perfectly matched DNA strands, which occur from the hybridization of complementary DNA hyperedges from training data for ‘6’ and ‘7’, the hybridization being when the negative weight term is calculated. In the case of the positive weight update, the resulting DNA concentrations of each hyperedge from cleavage and purification is amplified, where the positive term was also calculated from the initial hybridization process.”

We hope this clarifies this issue, but to add to this, we point out that the theoretical section of our paper is what is approximated by the molecular learning scheme which was designed primarily for implementation in vitro. In the current in vitro DNA design, the concentration of each DNA hyperedge strand is what represents the probabilistic weight of each hyperedge. The intensity of the fluorescent label resulting from the maximum concentration of DNA is used to determine the answer, which is different to in silico, where there is an exact integer value which is the representation of the output label.

Theoretically, updating the gradient with the positive and negative term as in Equation 3 can be calculated with the integer values, however, the experimentally approximated weight update entails the positive and negative weight update or increase and decrease of DNA concentrations based on the degree of matching between hybridized strands, as explained above. The relative DNA concentrations form the DNA dataset which approximates the probabilistic inference presented in Equation 1.

In regards to the elements of equation 1 and 3 and how these relate to the biochemical reaction rates or enzymatic rate:

Line 169-176

“The hybridization rate of DNA datasets to a) construct hyperedges, theta being the hyperdges made from the data and b) to calculate the positive and negative term of equation 3, is much faster due to the massively parallel nature of DNA computing, compared to the sequential matching of data in silico (Sections 3.1, 3.2 and Figures 6 and 7a). DNA data representation through the use of sticky ends and ligation enzyme is almost instantaneous too, due to the use of common complementary strands used to ligate single variable DNA to form free-order hyperedges. This step approximates the kernel function in Equation 1. The weight of hyperedges in silico is approximated by the relative concentrations of DNA hyperedges, the relative weights of DNA hyperedges being the probabilistic weight calculated in silico, and the updating of weights in Equation 3 occurs through the PCR amplification and S1 nuclease enzyme cleavage of DNA thereby increasing and decreasing the concentration of best matched DNA hyperedges respectively (Figure 7b).”

Due to the inevitable differences between the in silico and in vitro systems, we apologize that it seems an exact proof of what is occurring in the test tube and links with the theoretical components cannot be provided. However, we have tried to explain that each step of the systematic experimental protocol which approximates the equations to implement molecular machine learning is thoroughly written in the paper and we hope that this is sufficient to present our work to the readers of this special edition of the Molecules journal.

We have also uploaded an improved version of our code available on GitHub with a more comprehensive description of the code, explanation of different scripts and the Matlab toolboxes needed.

We sincerely thank you for your continued support and belief in our willingness to improve our work for submitting to this journal. All your comments were very much appreciated and we could not have completed this revised version without you!

Thank you.
